# Development and validation of a predictive model for postpartum endometritis

**Xiujuan Wang[1], Hui Shao [1]\*, Xueli Liu[2], Lili Feng[3]**

**1** Department of Infectology, Shaoxing Maternity and Child Health Care Hospital, Shaoxing, China,
**2** Department of Medical, Shaoxing Maternity and Child Health Care Hospital, Shaoxing, China,
**3** Department of Anesthesiology, Shaoxing Maternity and Child Health Care Hospital, Shaoxing, China

\* 15609631795@163.com

## Abstract

### Objective

The aim was to develop a predictive tool for anticipating postpartum endometritis occurrences and to devise strategies for prevention and control.

### Methods

Employing a retrospective approach, the baseline data of 200 women diagnosed with postpartum endometritis in a tertiary maternity hospital in Zhejiang Province, spanning from February 2020 to September 2022, was examined. Simultaneously, the baseline data of 1,000 women without endometritis during the same period were explored with a 1:5 ratio. Subsequently, 1,200 women were randomly allocated into a training group dataset and a test group dataset, adhering to a 7:3 split. The selection of risk factors for postpartum endometritis involved employing random forests, lasso regression, and traditional univariate and multifactor logistic regression on the training group dataset. A nomogram was then constructed based on these factors. The model's performance was assessed using the area under the curve (AUC), calculated through plotting the receiver operating characteristic (ROC) curve. Additionally, the Brier score was employed to evaluate the model with a calibration curve. To gauge the utility of the nomogram, a clinical impact curve (CIC) analysis was conducted. This comprehensive approach not only involved identifying risk factors but also included a visual representation (nomogram) and thorough evaluation metrics, ensuring a robust tool for predicting, preventing, and controlling postpartum endometritis.

### Results

In the multivariate analysis, six factors were identified as being associated with the occurrence of maternal endometritis in the postpartum period. These factors include the number of negative finger tests (OR: 1.159; 95%CI: 1.091–1.233; P < 0.05), postpartum hemorrhage (1.003; 1.002–1.005; P < 0.05), pre-eclampsia (9.769; 4.64–21.155; P < 0.05), maternity methods (2.083; 1.187–3.7; P < 0.001), prenatal reproductive tract culture (2.219; 1.411–3.47; P < 0.05), and uterine exploration (0.441; 0.233–0.803; P < 0.001). A nomogram was then constructed based on these factors, and its predictive performance was assessed

**Competing interests:** The authors have declared that no competing interests exist.

using the area under the curve (AUC). The results in both the training group data (AUC: 0.803) and the test group data (AUC: 0.788) demonstrated a good predictive value. The clinical impact curve (CIC) further highlighted the clinical utility of the nomogram.

## Conclusion

The development of an individualized nomogram for postpartum endometritis infection holds promise for doctors in screening high-risk women, enabling early intervention and ultimately reducing the rate of postpartum endometritis infection. This comprehensive approach, integrating key risk factors and predictive tools, enhances the potential for timely and targeted medical intervention.

## Introduction

Endometritis, an infectious and inflammatory disease affecting the endometrium, can be histo-pathologically classified into two distinct categories. Chronic endometritis is characterized by oedematous changes in the upper part of the endometrium, a high density of stromal cells, immaturity between the epithelium and stroma, and a reduction in the number of stromal plasma cells in the endometrial test. On the other hand, acute endometritis (AE) is defined by the formation of microabscesses and neutrophilic infiltration in the endometrial epithelium, glandular structures, and uterine cavities. Our study specifically focused on acute endometritis, presenting with symptoms such as fever, pelvic pain, and vaginal discharge within a few days after delivery [1]. AE is predominantly characterized by the presence of uterine pressure and pain, accompanied by an elevated temperature post-delivery. Notably, endometritis stands out as the most common postnatal infection, contributing to 27% of all postnatal complications [2]. Despite previous studies [3–5] identifying a limited set of independent factors associated with AE infection development in the postpartum period, the clinical application of this information remains constrained due to the lack of integration of these risk factors. There is a need for a more comprehensive approach that considers and combines these factors for a more effective clinical understanding and management of acute endometritis. If a straightforward predictive tool can enhance the accurate identification of independent risk factors for acute endometritis (AE) in the postpartum period, its integration into routine clinical practice holds the potential to guide effective prevention and control strategies. A nomogram, a tool that translates statistical equations into user-friendly graphs, serves as a reliable and convenient method for risk quantification. Despite the utility of nomograms in various medical contexts, their application for predicting the risk of developing endometritis in the postpartum period remains infrequently reported. Consequently, our study aimed to fill this gap by developing and validating a nomogram using easily accessible clinical data from hospitals, including admission characteristics, laboratory findings, and delivery details. Six independent risk factors for postpartum endometritis were ultimately identified through a comprehensive analysis, combining two machine learning approaches with traditional univariate and multifactorial logistic regression analyses. These findings were used to construct nomogram tailored for the swift identification of maternal postpartum patients potentially at high risk of developing postpartum endometritis. Implementing this measure will help mitigate the occurrence of acute postpartum endometritis, consequently alleviating both the physical discomfort and financial strain experienced by pregnant women. The outcome of our research is anticipated to offer a valuable reference point for clinical assessment regarding the incidence of AE. Moreover, it

has the potential to contribute to early prevention and control measures, thereby enhancing the overall management of AE in the postpartum period.

## Methodology

### Date of access to data

Data accessed on 1 December 2022 and authors had access to information that could identify individual participants during or after data collection.

### Type and medium of study

This retrospective, observational study took place at the Shaoxing Women's and Children's Hospital, a tertiary specialist hospital affiliated with the Shaoxing College of Arts and Sciences in East China. The hospital boasts a capacity of 600 beds and is equipped with six dedicated obstetrics wards, providing a total of 160 beds for maternal care.

### The study population

The study encompassed a cohort of 1077 women (Fig 1), comprising both mothers who developed AE and those who did not, following their delivery between February 2020 and September 2022 at our hospital. Inclusion criteria for patients were as follows: (1) Diagnosis of AE was determined based on the guidelines outlined in the Diagnosis of Hospital Infections by the National Health and Wellness Commission of China [6]. This involved the post-partum onset of symptoms such as fever or chills, lower abdominal pain or pressure, irregular vaginal bleeding, or foul-smelling malodor; and (2) inclusion required the availability of comprehensive information about the case. Exclusion criteria for patients were: (1) presence of intrauterine infections; (2) incomplete clinical information; and (3) women in critical condition.

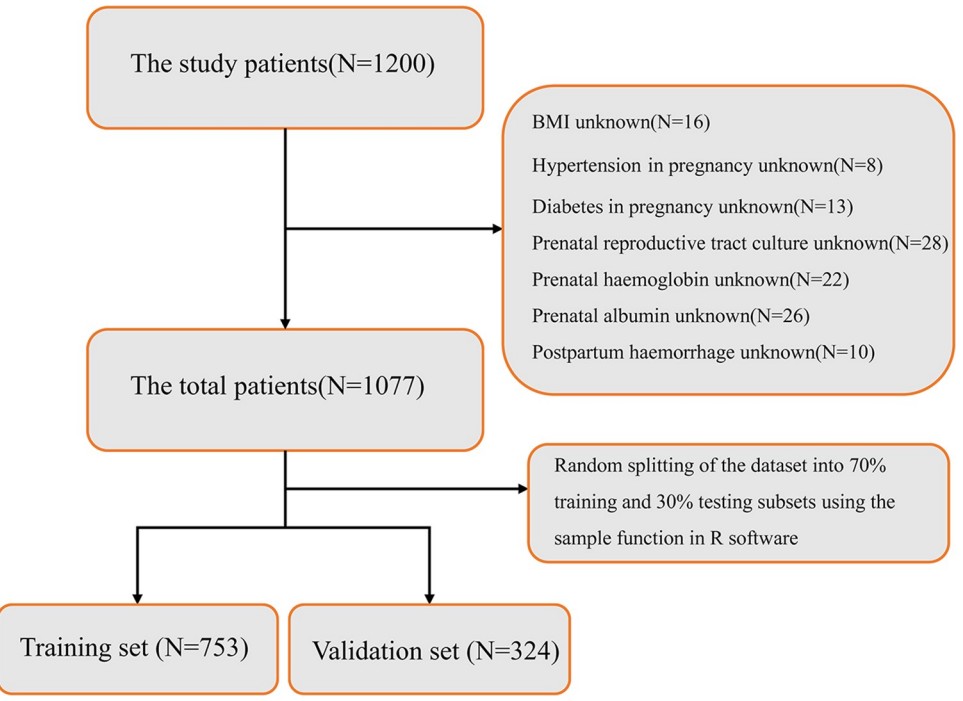

**Fig 1. Flow chart of patient selection.**

## Data collection

We conducted a retrospective compilation of clinical and laboratory data, as well as details of invasive procedures upon admission, drawing information from both the electronic medical record system and the nursing documentation system. The variables encompassed in this comprehensive analysis included age, BMI, education, gestational week, number of pregnancies, prenatal conditions such as hypertension, diabetes mellitus, and anemia, along with pre-eclampsia. Additionally, data on the mode of delivery, prenatal hospital stay, admission genital tract culture, blood glucose levels, prenatal hemoglobin, albumin levels, the frequency of vaginal finger tests before and after rupture of membranes, duration of rupture of membranes, method of rupture of membranes, whether labor was induced by a balloon, the extent of postpartum hemorrhage, and whether the uterus was explored were systematically gathered. The occurrence of AE was assessed by querying patients through our hospital infection surveillance system.

## Ethics statement

This retrospective study was approved by the Ethics Committee of Shaoxing Maternal and Child Health Hospital in January 2022, it did not involve animal or human clinical trials. And data collection was based on complete medical records and data analysis was performed anonymously. All our research methods were in accordance with relevant guidelines and regulations.

## Statistical analysis

The equality of distribution for continuous parameters was assessed using the Kolmogorov-Smirnov test. For normally distributed data, mean ± standard deviation was employed, while skewed data were represented using median and interquartile range. Univariate analyses involved the use of independent t-tests and Mann-Whitney U-tests to scrutinize differences in continuous variables between groups, with chi-square and Fisher's exact tests applied to categorical variables. We conducted collinearity diagnostics on the 19 factors included in the study. To identify the risk factors for AE, we employed random forest, lasso regression, as well as univariate and multivariate logistic analyses. The outcomes of these analyses yielded the risk factors essential for plotting the nomogram. The predictive performance of the nomogram underwent assessment through discrimination and calibration measures. ROC curves were employed for evaluating model discrimination [7], while Brier score and calibration plots were utilized for assessing model calibration. Clinical impact curves (CIC) were plotted to determine the clinical utility of the predicted column line plots, describing the net benefit of the model. Subgroup analyses were additionally conducted to enhance the robustness of the predictive models. A significance level of $P < 0.05$ was considered statistically significant. All analyses were performed using R software version 4.2.2, with the involvement of R packages such as "readr," "compareGroups," "foreign," "rms," "rmda," "pROC," "nomogramFormula," "forestmodel," "ggplot2," "RColorBrewer," "tidyverse," "randomForest," "rfPermute," "openxlsx," "jstable," "forestploter," "car," "grid."

## Results

### Population baseline characteristics

A total of 1,077 hospitalized women meeting the inclusion criteria participated in the study, with 194 (18.1%) experiencing postpartum AE infection. The participants were divided into a training cohort (n = 753) and a validation cohort (n = 324) at a ratio of 7:3. Table 1 presents a

**Table 1. Characteristics of patients in the training and validation cohorts.**

| Variable | Training Cohort | Validation Cohort | p |
|---|---|---|---|
| | N = 753 | N = 324 | |
| Age, years (mean, ±SD) | 29.5 (4.35) | 29.1 (4.15) | 0.204 |
| BMI(mean, ±SD) | 27.4 (3.45) | 27.4 (3.79) | 0.903 |
| Educational attainment, n (%) | | | 0.108 |
| Junior high below | 246 (32.7%) | 123 (38.0%) | |
| Junior high above | 507 (67.3%) | 201 (62.0%) | |
| Gestation period, weeks (mean, ±SD) | 39.1 (1.72) | 39.1 (1.70) | 0.761 |
| Number of pregnancies (mean, ±SD) | 2.11 (1.28) | 2.09 (1.23) | 0.811 |
| Hypertension in pregnancy, n (%) | | | 0.150 |
| Yes | 22 (2.92%) | 4 (1.23%) | |
| No | 731 (97.1%) | 320 (98.8%) | |
| Diabetes in pregnancy, n (%) | | | 0.777 |
| Yes | 97 (12.9%) | 39 (12.0%) | |
| No | 656 (87.1%) | 285 (88.0%) | |
| Pre-eclampsia, n (%) | | | 0.422 |
| Yes | 38 (5.05%) | 21 (6.48%) | |
| No | 715 (95.0%) | 303 (93.5%) | |
| Maternity methods, n (%) | | | 0.547 |
| Natural birth | 418 (55.5%) | 187 (57.7%) | |
| Cesarean section | 335 (44.5%) | 137 (42.3%) | |
| Length of prenatal stay, day (mean, ±SD) | 1.42 (1.58) | 1.46 (1.61) | 0.756 |
| Prenatal reproductive tract culture, n (%) | | | 0.777 |
| Germ-carrying | 182 (24.2%) | 75 (23.1%) | |
| Germ-free | 571 (75.8%) | 249 (76.9%) | |
| Prenatal haemoglobin(mean, ±SD) | 118 (11.6) | 117 (12.1) | 0.200 |
| Prenatal albumin(mean, ±SD) | 34.2 (2.46) | 34.1 (2.90) | 0.432 |
| No. of vaginal finger tests(mean, ±SD) | 5.55 (4.07) | 6.05 (4.41) | 0.080 |
| Duration of membrane breaking(mean, ±SD) | 385 (1029) | 446 (818) | 0.295 |
| Mode of rupture of membranes, n (%) | | | 0.165 |
| Not man-made | 279 (37.1%) | 105 (32.4%) | |
| Man-made | 474 (62.9%) | 219 (67.6%) | |
| Balloon induction of labour, n (%) | | | 0.653 |
| Yes | 18 (2.39%) | 10 (3.09%) | |
| No | 735 (97.6%) | 314 (96.9%) | |
| Postpartum haemorrhage(mean, ±SD) | 344 (179) | 331 (138) | 0.191 |
| Uterine exploration, n (%) | | | 0.683 |
| Yes | 187 (24.8%) | 85 (26.2%) | |
| No | 566 (75.2%) | 239 (73.8%) | |
| Endometritis, n (%) | | | 0.621 |
| Yes | 139 (18.5%) | 55 (17.0%) | |
| No | 614 (81.5%) | 269 (83.0%) | |

comparison of clinicopathological characteristics between the training and validation datasets. No significant differences in clinicopathological features were observed between the two cohorts (p > 0.05). Logistic one-way analysis of the training set, as illustrated in Table 2, identified 10 variables (p < 0.05) as risk factors associated with maternal postpartum AE infection. These variables include "BMI," "Number of pregnancies," "Pre-eclampsia," "Maternity

**Table 2. Risk factors for AE infections in the training cohort.**

| Variable | *Endometritis* | *Non-Endometritis* | *P* |
|---|---|---|---|
| | *N = 614* | *N = 139* | |
| Age, years (mean, ±SD) | 29.6 (4.33) | 29.1 (4.43) | *0.233* |
| BMI(mean, ±SD) | 27.3 (3.41) | 27.9 (3.60) | ***0.049*** |
| Educational attainment, *n* (%) | | | *0.223* |
| Junior high below | 194 (31.6%) | 52 (37.4%) | |
| Junior high above | 420 (68.4%) | 87 (62.6%) | |
| Gestation period, weeks (mean, ±SD) | 39.1 (1.56) | 38.8 (2.27) | *0.088* |
| Number of pregnancies (mean, ±SD) | 2.2 (1.29) | 1.9 (1.25) | ***0.011*** |
| Hypertension in pregnancy, *n* (%) | | | *0.055* |
| Yes | 14 (2.3%) | 8 (5.8%) | |
| No | 600 (97.7%) | 131 (94.2%) | |
| Diabetes in pregnancy, *n* (%) | | | *0.467* |
| Yes | 76 (12.4%) | 21 (15.1%) | |
| No | 538 (87.6%) | 118 (84.9%) | |
| Pre-eclampsia, *n* (%) | | | *<0.001* |
| Yes | 15 (2.4%) | 23 (16.5%) | |
| No | 599 (97.6%) | 116 (83.5%) | |
| Maternity methods, *n* (%) | | | ***0.003*** |
| Natural birth | 357(58.1%) | 61 (43.9%) | |
| Cesarean section | 257(41.9%) | 78 (56.1%) | |
| Length of prenatal stay, day (mean, ±SD) | 1.3 (1.37) | 1.9 (2.22) | *<0.001* |
| Prenatal reproductive tract culture, *n* (%) | | | *<0.001* |
| Germ-carrying | 130 (21.2%) | 52 (37.4%) | |
| Germ-free | 484 (78.8%) | 87 (62.6%) | |
| Prenatal haemoglobin(mean, ±SD) | 118.2 (11.16) | 117.0(13.24) | *0.265* |
| Prenatal albumin(mean, ±SD) | 34.3 (2.46) | 34.0 (2.47) | *0.316* |
| No. of vaginal finger tests(mean, ±SD) | 5.3 (3.98) | 6.7(4.26) | *<0.001* |
| Duration of membrane breaking(mean, ±SD) | 334.9 (594.2) | 603.7 (2036.4) | ***0.005*** |
| Mode of rupture of membranes, *n* (%) | | | *1.000* |
| Not man-made | 228 (37.1%) | 51 (36.7%) | |
| Man-made | 386 (62.9%) | 88(63.3%) | |
| Balloon induction of labour, *n* (%) | | | *0.181* |
| Yes | 12(2.0%) | 6 (4.3%) | |
| No | 602(98.0%) | 133 (95.7%) | |
| Postpartum haemorrhage(mean, ±SD) | 320.9 (116.93) | 444.5(319.02) | *<0.001* |
| Uterine exploration, *n* (%) | | | ***0.009*** |
| Yes | 165 (26.9%) | 22 (15.8%) | |
| No | 449(73.1%) | 117 (84.2%) | |

methods," "Length of prenatal stay," "Prenatal reproductive tract culture," "No. of vaginal finger tests," "Duration of membrane breaking," "Postpartum hemorrhage," and "Uterine exploration."

## Multivariate analysis

In the training cohort, we incorporated 19 clinically based indicators into the univariate analysis of endometritis infection. The analysis revealed that 10 variables (p < 0.05) may contribute to maternal postpartum infection with AE. Subsequently, these 10 variables underwent

**Table 3. Multivariate logistic analyses of risk factors for AE infections in the training cohort.**

| Variable | Estimate | Std.Error | OR(95%CI) | P-value |
|---|---|---|---|---|
| No. of vaginal finger tests | 0.1473 | 0.0312 | 1.16 (1.09–1.23) | 2.366884e-06 |
| Postpartum haemorrhage | 0.0033 | 0.0007 | 1.003 (1.002–1.005) | 1.274817e-06 |
| Pre-eclampsia | 2.2792 | 0.3847 | 9.77 (4.64–21.16) | 3.140495e-09 |
| Maternity methods | 0.7337 | 0.2895 | 2.08 (1.19–3.70) | 1.124706e-02 |
| Prenatal reproductive tract culture | 0.7965 | 0.2291 | 2.22 (1.41–3.47) | 5.060729e-04 |
| Uterine exploration | -0.8191 | 0.3141 | 0.44 (0.23–0.80) | 9.118538e-03 |

multivariate analysis, leading to the identification of six factors independently associated with AE infection through multifactorial logistic regression analysis (Table 3). These factors include "No. of vaginal finger tests," "Postpartum hemorrhage," "Pre-eclampsia," "Maternity methods," "Prenatal reproductive tract culture," and "Uterine exploration."

## Random forest and lasso regression analysis

Fig 2A illustrates the variable significance order for postpartum endometritis (AE) using the Random Forest algorithm in the training group. Variables marked with a * indicate significance ($p < 0.05$), with the order being "No. of vaginal finger tests," "Postpartum hemorrhage," "Pre-eclampsia," "Duration of membrane breaking," "Maternity methods," "Uterine exploration," and "Prenatal reproductive tract culture." When compared with the results of traditional single followed by multiple logistic regression analysis, only the variable "Duration of membrane breaking" was additionally identified. Fig 2B depicts a visualization of the relationship between the overall error rate and tree size. The black line represents the trend value of the Out-of-Bag (OOB) error rate, while the upper and lower lines denote the categorical error rate of the outcome variable. As the number of trees increases, the OOB error rate gradually decreases and stabilizes. Notably, there is a category of the outcome variable where the error rate initially decreases but then stabilizes, attributed to a significant class imbalance between

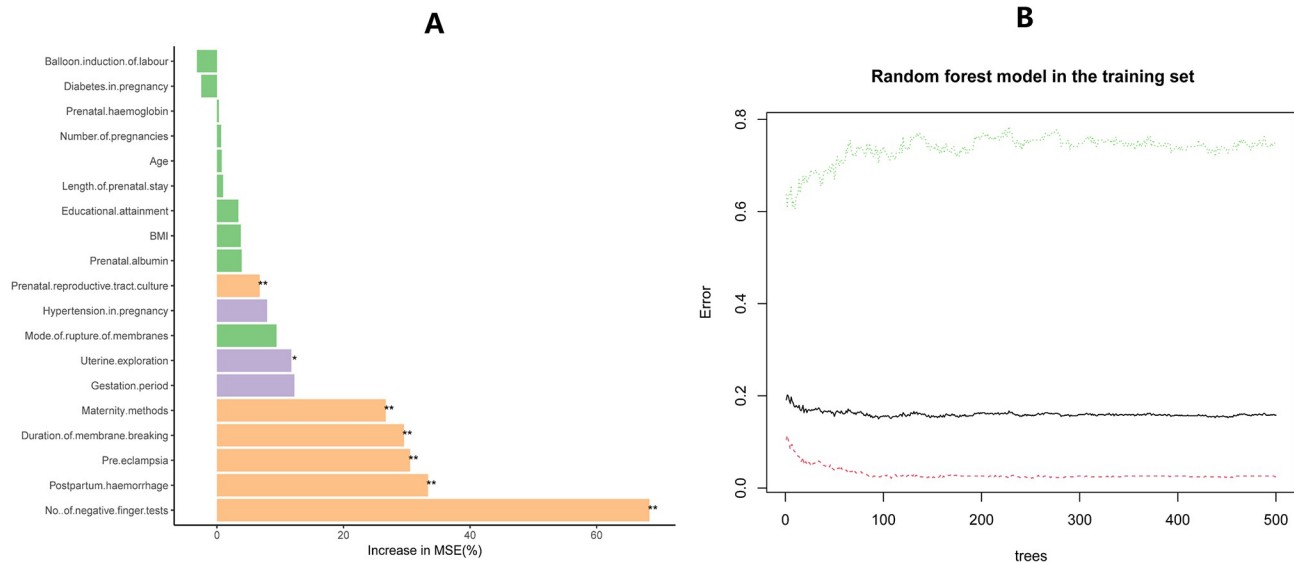

**Fig 2.** The significance and importance ranking of the 19 variables in the training group (A).a plot illustrating the relationship between tree size generated by the Random Forest algorithm and both the Out-of-Bag (OOB) error rate and classification error rate in the training group (B).

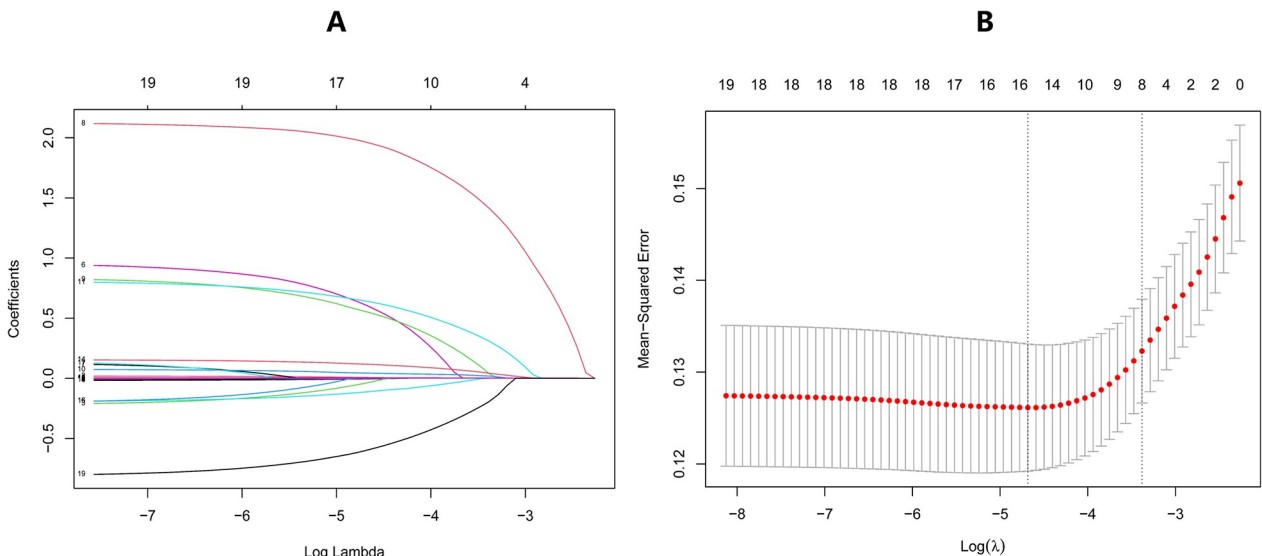

**Fig 3. Plot of the results of cross-validation, and the red dots in the figure represent the target parameters corresponding to each lambda (A).** 16 variables screened from 19 based on LASSO regression. A coefficient profile plot was produced against the log (λ) sequence.

the two outcome categories (approximately 1:4.5). Fig 4B displays the ROC plot of the random forest prediction model in the training set, exhibiting an area under the curve of 0.807. This value closely approximates the area under the curve of 0.803 observed for the prediction model constructed using traditional logistic regression. Fig 3A and 3B display the lasso regression screening predictors for the training group, identifying a total of 16 predictors. These predictors were subsequently included in the multifactorial logistic regression analysis (Table 4), resulting in the screening of 6 predictors: "No. of vaginal finger tests," "Postpartum hemorrhage," "Pre-eclampsia," "Maternity methods," "Prenatal reproductive tract culture," and "Uterine exploration." The outcomes aligned with those of the traditional single followed by

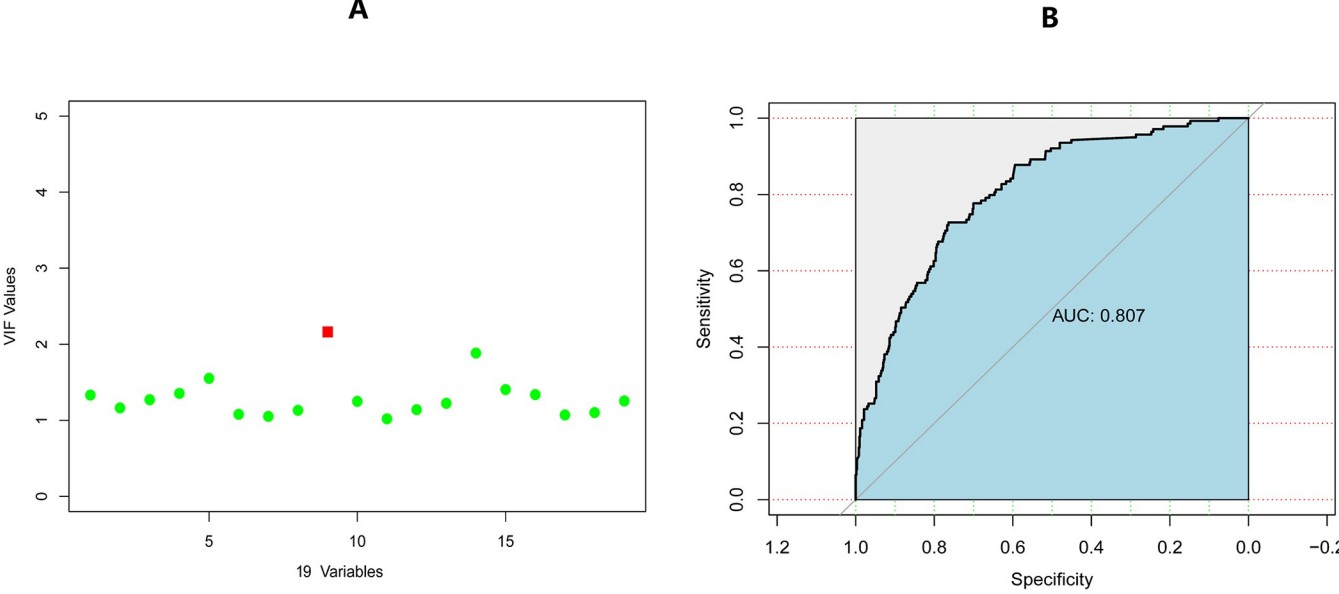

**Fig 4.** Plot of variance inflation factor (VIF) for 19 variables (A). ROC curves based on the Random Forest algorithm in the training group.

**Table 4. Multivariate regression model based on LASSO regression results.**

| Vriable | Estimate | Std.Error | Z-value | OR(95%CI) | P-value |
|---|---|---|---|---|---|
| No.ofvaginal finger tests | 1.562e-01 | 3.250e-02 | 4.807 | 1.70 (1.10–1.25) | 1.53e-06 |
| Postpartum hemorrhage | 3.241e-03 | 6.737e-04 | 4.811 | 1.00 (1.00–1.01) | 1.50e-06 |
| Pre-eclampsia | 2.173e+00 | 4.058e-01 | 5.355 | 8.78 (4.00–19.82) | 8.58e-08 |
| Maternity methods | 8.475e-01 | 3.017e-01 | 2.809 | 2.33 (1.30–4.25) | 4.97 e-03 |
| Prenatal reproductive tract culture | 7.981e-01 | 2.308e-01 | 3.458 | 2.22 (1.41–3.49) | 5.44 e-04 |
| Uterine exploration | -8.000e-01 | 3.157e-01 | -2.534 | 0.45 (0.24–0.82) | 1.13 e-02 |

multiple logistic regression analysis. Consequently, we opted for single-factor followed by multifactor logistic regression to construct the prediction model.

## Tests for covariance of variables

Following a general principle, a variance inflation factor (VIF) exceeding 2 suggests some degree of covariance, while a VIF surpassing 5 or 10 signifies a significant covariance issue. Fig 4A reveals that the VIF values for all 19 variables are below 2, indicating an absence of covariance among them.

## Internal evaluation

Six variables identified through multivariate analysis were utilized in constructing a predictive nomogram (Fig 5) for forecasting the occurrence of postpartum AE using R 4.2.2. Each

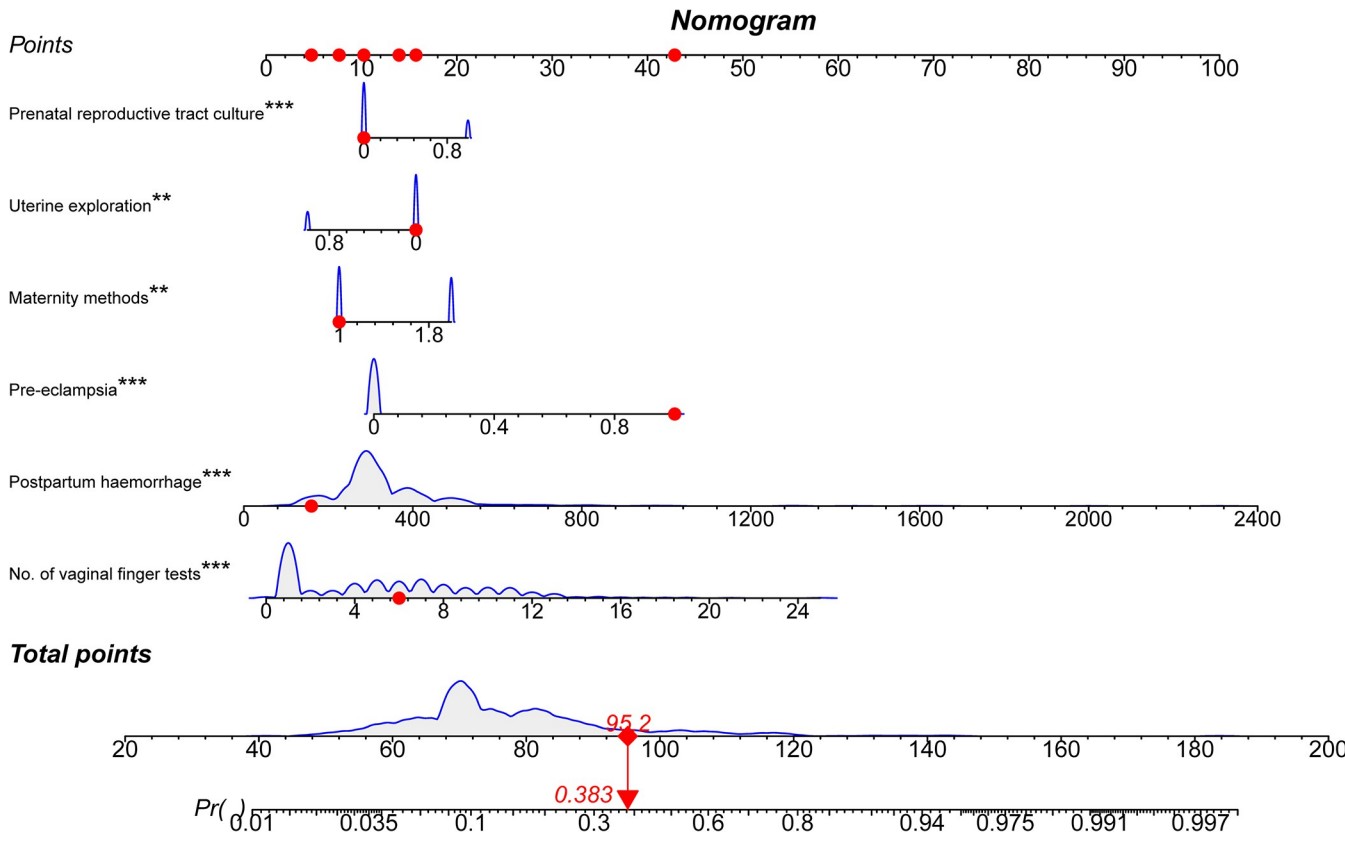

**Fig 5. Nomogram for prediction of maternal AE infections in women.**

variable was associated with specific points by drawing a line vertically to the points axis. Once the sum of these points was pinpointed on the total points axis, a line was drawn downward to the probability axis, indicating the probability of endometritis infection. The patient's score on the nomogram, with a total of 200 points, is illustrated by the red dot in the graph. In this case, the third-row observation in the training cohort corresponds to a woman admitted to the hospital as a carrier of a positive genital tract culture. She underwent no intrauterine exploration after delivery, had a spontaneous delivery, experienced preeclampsia, and had a labor hemorrhage of 160 ml. The total score amounted to 95.2, resulting in a calculated probability of endometritis infection at 0.383.

## Model performance evaluation

Fig 7A presents the calibration curve of the regression model within the training cohort, showcasing the excellent performance of both the calibration curve and Brier score (0.121 < 0.25). The Brier score assesses the overall performance of a model, ranging from 0 to 1. A score close to 0 signifies near-perfect model performance, indicating close alignment between predicted and actual values. Conversely, a Brier score > 0.25 suggests significant discrepancies between predictions and actual outcomes, rendering the model less meaningful. In essence, higher scores indicate poorer model effectiveness. The Area Under the Curve (AUC) of 0.803 (95% CI, 0.764–0.842) signifies the well-differentiated nature of the model in the training cohort [Fig 6A]. Validation of the calibration plots for the regression model, specifically for endometritis infection, was conducted using the validation cohort [Fig 7B]. The Brier score (0.11 < 0.25) suggests a minimal difference between predicted and actual values. In the validation group, the AUC [Fig 6B] was 0.788 (95% CI, 0.715–0.861), indicating the model's effective differentiation in both the training and validation cohorts.

## Clinical use of nomogram

Utilizing the ROC curve for model evaluation acknowledges the inevitability of false positives or negatives in real clinical settings. In response, we crafted a Clinical Impact Curve (CIC) based on 1077 maternity cases. The CIC analysis illuminates the clinical efficiency of our

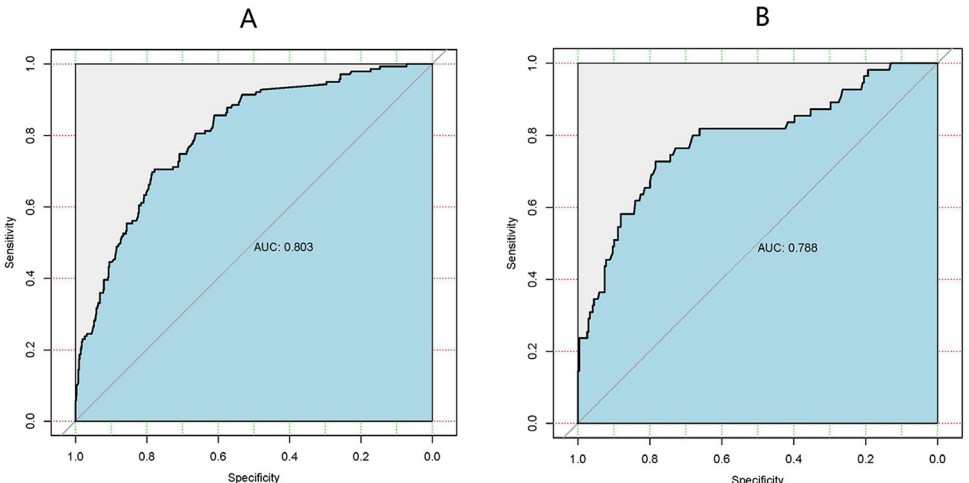

**Fig 6. Goodness of fit of the predicted risk and actual risk of AE infections.** (A) The ROC curves of the model in training sets. (B) the ROC curves of the model in validation sets. ROC curves depict discrimination capability of nomogram model. The larger the area of the AUC, the higher the prediction accuracy of the model.

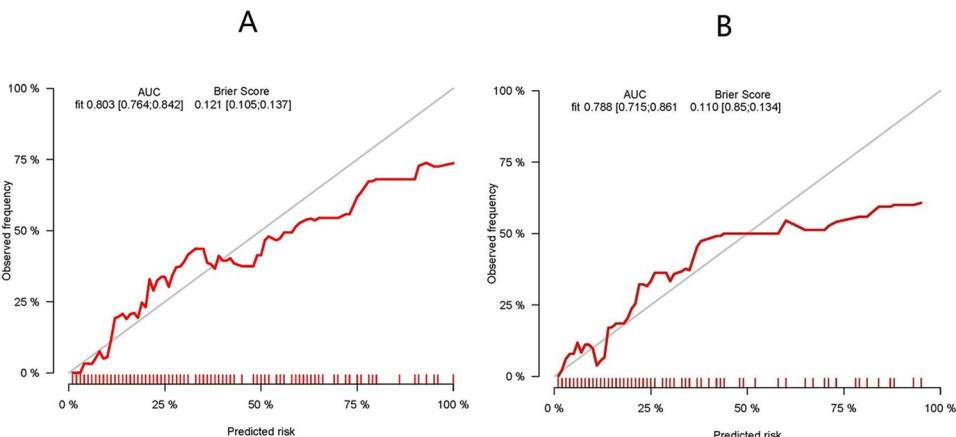

**Fig 7. Goodness of fit of the predicted risk and actual risk of AE infections.** (A) Calibration curves of the model in the training set. (B) Calibration curves of the model in the validation set. The 45-degree long dotted line represents a perfect prediction, and the solid line represent the predictive performance of the model. The closer the long dotted line fit is to the ideal line, the better the predictive accuracy of the model.

predictive model. The red curve (Number high risk) depicts the count of individuals identified as positive by the model at each threshold probability (Number high risk). Simultaneously, the blue curve (Number high risk with event) illustrates the true positives at each threshold probability. Notably, when the threshold probability exceeds 55% of the predictive score probability value, the model identifies the AE high-risk population, aligning closely with the actual AE population. This affirmation establishes the clinical efficiency of our predictive model, visually showcasing the net benefit of the nomogram (Fig 8).

To enhance the nomogram's practicality in clinical settings, we systematically explored various critical values aligned with interpretability based on our accumulated knowledge and practical insights. The total score for each patient in the training cohort was computed using the nomogram, and subsequently, the patient scores were stratified into four distinct

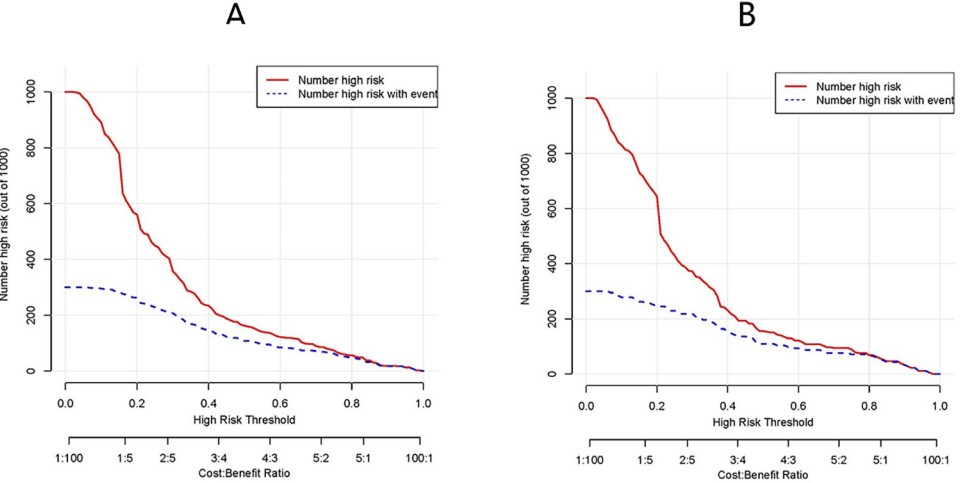

**Fig 8.** (A)The clinical impact curve(CIC) analysis of the nomogram in the training set. (B)The clinical impact curve (CIC) analysis of the nomogram in the validation set. The red line (number of high-risk individuals) is the number of people who are classified as positive by the model at each threshold probability, the blue line (number of high-risk individuals with event) indicates the number of true positives at each threshold probability.

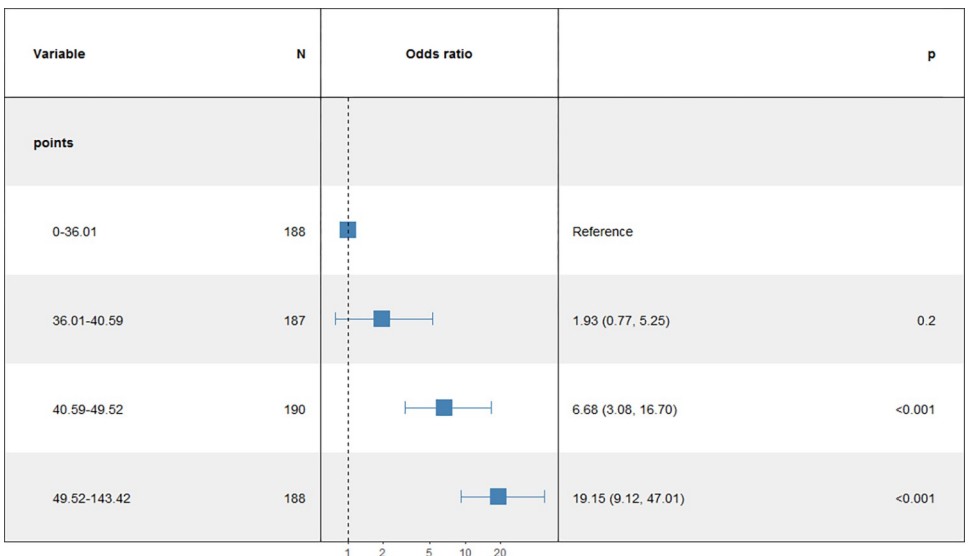

**Fig 9. Association between the total points of the nomogram and AE infections.**

subgroups based on quartiles. These subgroups were then incorporated into a logistic regression model to assess the risk of endometritis infection across different risk categories. As illustrated in Fig 9, a positive correlation was observed between the total score and the risk of coronary heart disease. Participants in the upper quartile (total score: 49.52–143.42) exhibited a significantly higher risk of coronary heart disease compared to those in the lower quartiles (0–36.02), (36.02–40.59), and (40.59–49.52) (odds ratio [OR]: 19.15, 95% confidence interval [CI]: 9.12–47.01). Consequently, we identified four risk subgroups: low-risk (≤36.02), lower-risk (36.02 < Total points ≤40.59), higher-risk (40.59 < Total points < 49.52), and high-risk (≥49.52).The predictive model demonstrated effective discrimination between high-risk and low-risk groups, providing a valuable foundation for preventive strategies against postpartum AE infection in pregnant women.

## Subgroup analysis

To enhance the validation of our prediction model, we performed two subgroup analyses: one based on age (Fig 10A) and another on pregnancy status (Fig 10B). The analysis results indicated that the p-value for interaction among each factor exceeded 0.05. This suggests the absence of any subgroup effects, indicating that each factor can be employed to evaluate the risk level of postpartum acute endometritis occurrence across different age groups and number of pregnancies in pregnant women. These findings further reinforce the reliability and strength of the prediction model.

## Discussion

Postpartum infections stand as a primary contributor to postpartum morbidity, often leading to extended hospital stays and readmissions. Moreover, these infections exert a substantial financial burden on both families and society at large, with endometritis emerging as a prevalent postpartum infection [2]. Within the scope of this retrospective study involving 1077 laboring women, the findings highlighted several factors associated with an increased risk of maternal postpartum AE infection. These factors include "No. of vaginal finger tests," "Postpartum haemorrhage," "Pre-eclampsia," "Maternity methods," "Prenatal reproductive tract

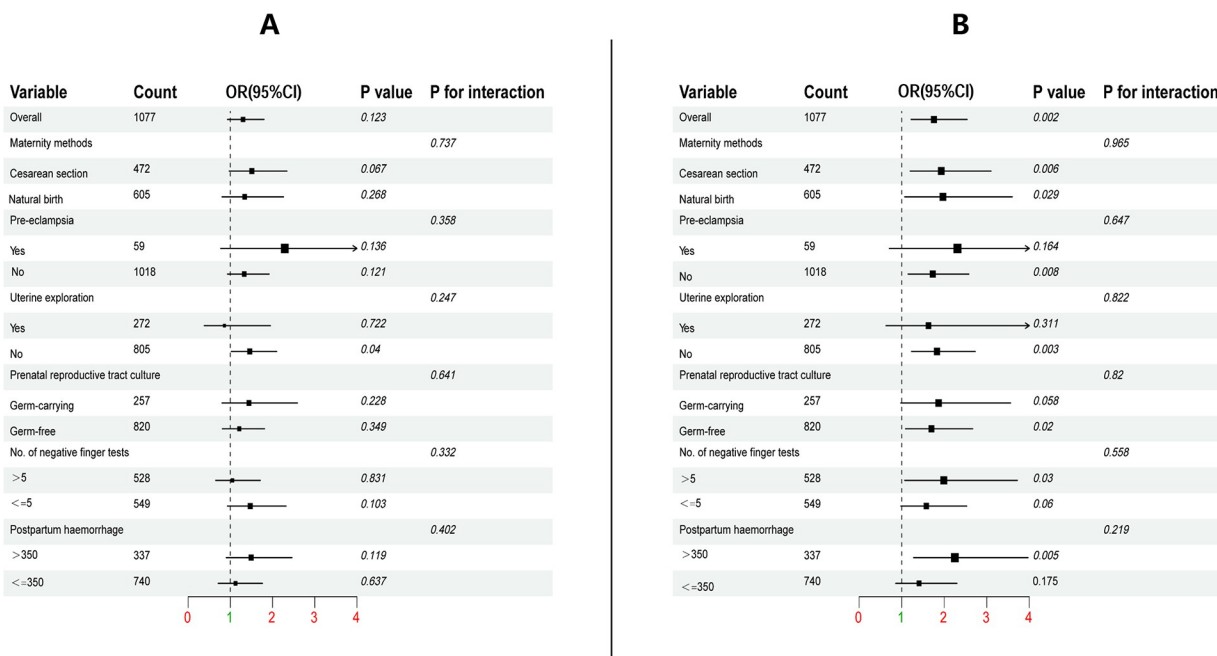

**Fig 10.** Subgroup analyses were conducted by dividing the participants into two groups based on age, with a cut-off of 30 years(A). Subgroup analyses were performed by categorizing the participants into two groups based on the number of pregnancies, with a cut-off of 2(B).

culture," "Uterine reproductive tract culture," "Uterus," and "Uterine exploration." Each of these elements contributes as a risk factor in the context of postpartum AE infection, emphasizing the need for comprehensive preventive measures and interventions in maternal care.

The quantity of negative finger tests emerges as a significant risk factor for postpartum AE infection, as evidenced by a study revealing an increased risk of maternal infection associated with cervical finger tests [8]. The transvaginal finger test involves meticulous steps, including vulva disinfection, the use of sterile gloves by the examiner, and direct contact with the pelvis and birth canal using the index and middle fingers. This procedure aims to assess the condition of the pelvis and birth canal, comprehend cervical canal regression, and evaluate the dilatation of the uterine opening. Additionally, it provides insights into the height of prenatal dew, fetal orientation, and the presence or absence of the umbilical cord beneath the prenatal dew. The examiner also conducts a Bishop's score to assess cervical ripeness. In theory, the normal vagina harbors diverse microbial populations, and their relationship with the host vagina is typically interdependent, maintaining a dynamic ecological balance under normal circumstances. However, the equilibrium of vaginal microecology is disrupted by multiple vaginal palpation examinations. This disruption, coupled with the potential damage to the vaginal mucosa during these examinations, creates conditions favorable for the invasion and upward movement of pathogenic bacteria. Consequently, this can lead to violations of the endometrium, and in severe cases, the myometrium, resulting in infections [9, 10]. Prior research has established a connection between reproductive tract infections and histologically confirmed endometritis [11]. These infections, when occurring during pregnancy, can disrupt the hormonal balance in the body, leading to an alteration in vaginal flora and a reduction in mucosal barrier function. The toxins produced by invading pathogenic bacteria, such as mycoplasma, are particularly damaging to membranes, increasing the likelihood of premature rupture. This, in turn, escalates the risk of uterine cavity infections and puerperal infections. Notably, preoperative vaginal disinfection for mothers undergoing a cesarean section significantly reduces

the incidence of endometritis post-delivery [12–14]. A specific study highlighted a 17-fold higher risk of wound infection in women delivering via cesarean section compared to those undergoing vaginal non-instrumental delivery. The risk of endometritis was also elevated in the case of cesarean deliveries [15]. The trauma and mechanical stimulation associated with cesarean sections can result in decreased maternal immunity. Furthermore, surgical procedures induce changes in the ecological environment of the reproductive tract, characterized by a decline in anaerobic bacteria and an increase in aerobic bacteria. There's also a risk of peritoneal contamination with bacteria from the uterine cavity [16, 17]. Additionally, inadequate management of the surgical incision may lead to delayed tissue healing, further amplifying the risk of postpartum AE infection, particularly in primiparous women. Blood serves as a rich culture medium for pathogenic bacteria. In cases where there is a substantial amount of bleeding during delivery, it essentially creates favorable conditions for the proliferation of pathogenic bacteria. This heightened bacterial presence elevates the risk of maternal puerperal infection. Additionally, patients with hypertensive disorders in pregnancy experience disrupted hemodynamics in the endometrium, coupled with higher blood pressure. This makes them more susceptible to postnatal hemorrhage, further increasing the likelihood of postpartum infections [18]. The varying degrees of blood loss during childbirth, surgery, and postpartum hemorrhage contribute to an imbalance in the mother's nutritional status. This leads to a decline in hemoglobin and albumin levels, resulting in malnutrition, anemia, and a compromised immune system. These factors collectively amplify the risk of postpartum endometritis infection in the mother. Previous research has indicated that preeclampsia plays a crucial role in influencing maternal endometritis infection [19]. The systemic spasm of small vessels in patients with pre-eclampsia induces local tissue ischemia and hypoxia, creating an environment conducive to the rapid multiplication of anaerobic bacteria. These anaerobic bacteria, in turn, generate gas, elevating local tissue tension while diminishing blood supply, thereby promoting infection development. Moreover, individuals with preeclampsia often exhibit higher blood pressure and a susceptibility to post-partum hemorrhage, further escalating the risk of postpartum infection. Conducting uterine cavity exploration post-delivery proves beneficial in minimizing infection incidence. During delivery, remnants of fetal membranes and placenta can persist in the uterine cavity, impacting uterine contraction and increasing the likelihood of bleeding. If not promptly addressed, necrotic tissue forms, becoming a breeding ground for pathogenic bacteria. Ensuring uterine cavity exploration adheres strictly to aseptic procedures and promptly removing residual tissues contributes to reducing infection incidence. Research has demonstrated [20] that intrauterine cleansing promptly eliminates malodors, thereby reducing the risk of postpartum endometritis infection in mothers.

Certainly, our study is not without limitations. Firstly, being a retrospective study, there exists a potential for selection bias. Secondly, the fact that it was conducted at a single center highlights the necessity for external validation. The high incidence of postpartum acute endometritis (AE) is indeed concerning. Therefore, in the future, we aim to extend the validation of our findings beyond our current scope, encompassing other regions within the province and potentially the entire country. This endeavor will likely entail collaboration with numerous tertiary specialized maternity and child hospitals. Furthermore, we are contemplating the initiation of a prospective cohort study to formulate more comprehensive guidelines for the prevention of postpartum AE. It's worth noting that our study only utilized two machine learning methods. Considering that several studies [21–23] have employed more sophisticated and diverse machine learning techniques to construct and evaluate predictive models, followed by performance comparisons to identify the most effective method, we recognize the value in adopting a similar approach. This could lead to a deeper understanding of the predictive modeling landscape and aid in selecting the optimal machine learning method for model

construction. In conclusion, our findings indicate that six independent risk factors are linked to maternal endometritis infection during the postpartum period. These predictors are not only easily accessible in clinical practice but also elucidate the clinical significance of the model. Implementing measures to control for these factors in clinical practice has the potential to mitigate maternal postpartum AE infections.

## Conclusions

This study revealed that independent risk factors for postpartum endometritis infection include the absence of intrauterine cleansing, elevated postpartum hemorrhage, caesarean section, positive genital tract culture upon admission, and a high number of vaginal fingerprinting instances. A nomogram, constructed from these identified risk factors, boasts a minimal number of variables and user-friendly design, demonstrating robust predictive power. Such a nomogram holds significant potential in guiding clinical decision-making and aiding in treatment planning for postpartum endometritis infections.

## Supporting information

**S1 Dataset.**
(CSV)

**S1 File.**
(DOCX)

## Author Contributions

**Data curation:** Xueli Liu, Lili Feng.

**Methodology:** Xiujuan Wang.

**Writing – original draft:** Xiujuan Wang, Xueli Liu.

**Writing – review & editing:** Hui Shao.

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
