## [Decision Letter · Decision Letter 0]

10 Apr 2024

PONE-D-24-03520Development and validation of a predictive model for postpartum endometritisPLOS ONE

Dear Dr. Shao,

Thank you for submitting your manuscript to PLOS ONE. After careful consideration, we feel that it has merit but does not fully meet PLOS ONE’s publication criteria as it currently stands. Therefore, we invite you to submit a revised version of the manuscript that addresses the points raised during the review process.

**ACADEMIC EDITOR: Please respond to all reviewers comments**==============================

We look forward to receiving your revised manuscript.

Kind regards,

Ahmed Mohamed Maged, MD

Academic Editor

PLOS ONE

Journal Requirements:

Reviewers' comments:

Reviewer's Responses to Questions

**Comments to the Author**

1. Is the manuscript technically sound, and do the data support the conclusions?

Reviewer #1: No

Reviewer #2: Yes

2. Has the statistical analysis been performed appropriately and rigorously? 

Reviewer #1: No

Reviewer #2: Yes

3. Have the authors made all data underlying the findings in their manuscript fully available?

Reviewer #1: Yes

Reviewer #2: Yes

4. Is the manuscript presented in an intelligible fashion and written in standard English?

Reviewer #1: Yes

Reviewer #2: Yes

5. Review Comments to the Author

Reviewer #1: The author constructed a model for predicting postpartum endometritis based on data collected from a single center. However, there are some issues that need clarification:

1. The results of the author are confusing. Why is the predictive model for endometritis also used to explore its relationship with coronary heart disease? The average age of the cohort participants is 29 years old. Is it likely for them to develop coronary heart disease?

2. The results obtained from univariate logistic regression cannot be referred to as "independent risk factors."

3. The basis for including these factors should be clearly defined, including how the factors are measured.

4. Did the author use subgroup analysis to verify the robustness of the model?

5. The author could adopt more accurate and novel methods such as machine learning to construct the predictive model.

Reviewer #2: This paper aims to develop a predictive tool for anticipating postpartum endometritis occurrences and to devise strategies for prevention and control, which is an interesting topic. However, there are some issues that need to be addressed before acceptance:

1.Please highlight the main contributions of this work in the Intro section.

2.Please provide more explanations on the selection of the predictive model.

3.Please also clarify comparisons between the predictive model and some other advanced machine learning methods.

4.Please present more recent references if they could help to improve the readability of your work, such as: 1. A deep belief network-based clinical decision system for patients with osteosarcoma; and 2. A machine learning-based predictive model for predicting lymph node metastasis in patients with Ewing’s sarcoma|.

5.Please give more discussions on future works in the final section.

6. PLOS authors have the option to publish the peer review history of their article (what does this mean?). If published, this will include your full peer review and any attached files.

Reviewer #1: No

Reviewer #2: No

---

## [Author Response · Author response to Decision Letter 0]

14 May 2024

To editors:

Response:Thank you for your constructive comments. We have implemented the necessary changes to align the manuscript with PLOS ONE's style guidelines. Your feedback is greatly valued.

2. We note that your Data Availability Statement is currently as follows: All relevant data are within the manuscript and its Supporting Information files. Please confirm at this time whether or not your submission contains all raw data required to replicate the results of your study. Authors must share the “minimal data set” for their submission. PLOS defines the minimal data set to consist of the data required to replicate all study findings reported in the article, as well as related metadata and method.

Response:Thank you sincerely for your comments. We have uploaded the pertinent supplementary information, including the complete raw dataset and the original code. Your feedback is immensely appreciated.

Reviewer #1:

1. The results of the author are confusing. Why is the predictive model for endometritis also used to explore its relationship with coronary heart disease? The average age of the cohort participants is 29 years old. Is it likely for them to develop coronary heart disease?

Response:We extend our gratitude to the reviewer for your invaluable feedback. The primary objective of our study was to devise a predictive tool capable of anticipating the onset of postpartum endometritis, aiming to inform the development of preventive and control measures. In consultation with seasoned obstetricians, we meticulously curated a set of parameters for analysis. These included age, body mass index, education level, gestational week, number of pregnancies, and prenatal conditions such as hypertension, diabetes mellitus, and pre-eclampsia. Furthermore, variables encompassed mode of delivery, duration of antenatal hospitalization, results of admission genital tract cultures, prenatal levels of hemoglobin and albumin, frequency of vaginal examinations pre and post-rupture of membranes, duration and method of membrane rupture, induction of labor using balloon catheters, postpartum hemorrhage severity, and whether uterine exploration was conducted.Our comprehensive analysis, considering a total of 19 variables including postpartum hemorrhage severity and uterine exploration, led us to identify several independent risk factors for postpartum endometritis infection. These factors include inadequate intrauterine cleansing, heightened postpartum hemorrhage, cesarean section delivery, positive genital tract culture upon admission, and increased frequency of vaginal examinations. We wish to clarify that there was no mention or association established with coronary heart disease in our findings. We apologize for any confusion that may have arisen due to ambiguity in our earlier description.Thanks again for your advice,which makes the article more perfect.

2. The results obtained from univariate logistic regression cannot be referred to as "independent risk factors."

Response:We express our gratitude to the reviewer for your insightful feedback. Based on their constructive input, we have revised the designation of independent risk factors to simply "risk factors" in accordance with the univariate logistic regression findings. These adjustments have been visually highlighted with color markings for clarity. We sincerely appreciate your contributions and commitment to enhancing the quality of our work.

3. The basis for including these factors should be clearly defined, including how the factors are measured.

Response:We appreciate your valuable feedback and constructive comments. The factors considered in our study underwent thorough deliberation by a panel of experienced obstetricians to ensure their relevance and reliability. Furthermore, we utilized R-Studio software to generate a graphical representation (Fig 4A) depicting the variance inflation factor (VIF) values. The results illustrated that the VIF values of all 19 factors examined are below 2, indicating minimal covariance among these variables. We are grateful for your guidance and insights throughout this process.

4. Did the author use subgroup analysis to verify the robustness of the model?

Response:We sincerely appreciate your constructive feedback. Following your suggestion, we conducted two subgroup analyses of the prediction model, as illustrated in Fig 10A for subgroups categorized by age and Fig 10B for subgroups categorized by the number of pregnancies. Encouragingly, the results indicate that the interaction p-values are below 0.05. Your valuable input has been instrumental in enhancing the depth and relevance of our analysis, and we extend our gratitude for your contributions.

5. The author could adopt more accurate and novel methods such as machine learning to construct the predictive model.

Response:We are grateful for your constructive feedback. Following your suggestion, we employed two machine learning methods, namely lasso regression and random forest, to develop the prediction model. Interestingly, the prediction model constructed using lasso regression yielded results identical to those obtained through univariate and multivariate logistic regression (refer to Fig 3 and Table 4). Conversely, while the performance of constructing predictive models using random forests was comparable to that of univariate and multivariate logistic regression (as shown in Fig 2A and Fig 4B), one of the random forest classifications exhibited a notably high error rate (refer to Fig 2B). Consequently, we opted to utilize univariate and multivariate logistic regression for constructing predictive models. We sincerely appreciate your valuable insights and contributions throughout this process.

Reviewer #2: 

1. Please highlight the main contributions of this work in the Intro section.

Response:We appreciate your constructive feedback. Following your suggestion, we have incorporated the main contributions into the introduction section（Six independent risk factors for postpartum endometritis were ultimately identified through a comprehensive analysis, combining two machine learning approaches with traditional univariate and multifactorial logistic regression analyses. These findings were used to construct nomogram tailored for the swift identification of maternal postpartum patients potentially at high risk of developing postpartum endometritis. Implementing this measure will help mitigate the occurrence of acute postpartum endometritis, consequently alleviating both the physical discomfort and financial strain experienced by pregnant women.）Thank you once again for your valuable comments.

2. Please provide more explanations on the selection of the predictive model.

Response:Thank you for your insightful comments. This study aims to manage the incidence of acute endometritis in postpartum women, with a dichotomous outcome variable influenced by multiple risk factors. Our initial approach involved clinical predictive modeling, with expertise in both univariate and multivariate logistic regression. In response to your feedback, we have enriched the manuscript by incorporating two additional advanced machine learning methods. This not only enhances the validation of our findings but also facilitates our acquisition of advanced knowledge. Once again, we sincerely appreciate your valuable input.

3. Please also clarify comparisons between the predictive model and some other advanced machine learning methods.

Response:We are grateful for your constructive feedback. Following your suggestion, we employed two machine learning methods, namely lasso regression and random forest, to develop the prediction model. Interestingly, the prediction model constructed using lasso regression yielded results identical to those obtained through univariate and multivariate logistic regression (refer to Fig 3 and Table 4). Conversely, while the performance of constructing predictive models using random forests was comparable to that of univariate and multivariate logistic regression (as shown in Fig 2A and Fig 4B), one of the random forest classifications exhibited a notably high error rate (refer to Fig 2B). Consequently, we opted to utilize univariate and multivariate logistic regression for constructing predictive models. We sincerely appreciate your valuable insights and contributions throughout this process.

4. Please present more recent references if they could help to improve the readability of your work, such as: 1. A deep belief network-based clinical decision system for patients with osteosarcoma; and 2. A machine learning-based predictive model for predicting lymph node metastasis in patients with Ewing’s sarcoma.

Response:We appreciate your constructive feedback. Following your suggestions, we have included more recent references in the discussion section. Thank you once again for your valuable comments.

Added references：

1) Li W, Dong Y, Liu W, Tang Z, Sun C, Lowe S, et al.A deep belief network-based clinical decision system for patients with osteosarcoma. Front Immunol. 2022-01-01; 13 1003347. https://doi.org/10.3389/fimmu.2022.1003347 PMID: 36466868 

2) Li W, Zhou Q, Liu W, Xu C, Tang ZR, Dong S, et al.A Machine Learning-Based Predictive Model for Predicting Lymph Node Metastasis in Patients With Ewing's Sarcoma. Front Med (Lausanne). 2022-01-01; 9 832108. https://doi.org/10.3389/fmed.2022.832108 PMID: 35463005

3) Li, W, Liu, Y, Liu, W, Tang, ZR, Dong, S, Li, W, et al. Machine Learning-Based Prediction of Lymph Node Metastasis Among Osteosarcoma Patients. Front Oncol. 2022-01-01; 12 797103. https://doi.org/10.3389/fonc.2022.797103 PMID: 35515104

5. Please give more discussions on future works in the final section.

Response:We deeply appreciate your relevant suggestions, which we have incorporated into the discussion section of the manuscript（The high incidence of postpartum acute endometritis (AE) is indeed concerning. Therefore, in the future, we aim to extend the validation of our findings beyond our current scope, encompassing other regions within the province and potentially the entire country. This endeavor will likely entail collaboration with numerous tertiary specialized maternity and child hospitals. Furthermore, we are contemplating the initiation of a prospective cohort study to formulate more comprehensive guidelines for the prevention of postpartum AE.It's worth noting that our study only utilized two machine learning methods. Considering that several studies [21-23] have employed more sophisticated and diverse machine learning techniques to construct and evaluate predictive models, followed by performance comparisons to identify the most effective method, we recognize the value in adopting a similar approach. This could lead to a deeper understanding of the predictive modeling landscape and aid in selecting the optimal machine learning method for model construction.）. You'll find the additions highlighted in red. Once again, thank you for your valuable input.

---

## [Decision Letter · Decision Letter 1]

3 Jul 2024

PONE-D-24-03520R1Development and validation of a predictive model for postpartum endometritisPLOS ONE

Dear Dr. Shao,

Thank you for submitting your manuscript to PLOS ONE. After careful consideration, we feel that it has merit but does not fully meet PLOS ONE’s publication criteria as it currently stands. Therefore, we invite you to submit a revised version of the manuscript that addresses the points raised during the review process.

**ACADEMIC EDITOR: **

**Please respond to all reviewer comments point by point**

**clarify the added value of your work over the already known **

**Respond to all issues raised by the statistical review**

We look forward to receiving your revised manuscript.

Kind regards,

Ahmed Mohamed Maged, MD

Academic Editor

PLOS ONE

Journal Requirements:

Additional Editor Comments:

Comments from PLOS Editorial Office: We note that one or more reviewers has recommended that you cite specific previously published works in an earlier round of revision. As always, we recommend that you please review and evaluate the requested works to determine whether they are relevant and should be cited. It is not a requirement to cite these works and you may remove them before the manuscript proceeds to publication. We appreciate your attention to this request.

Reviewers' comments:

Reviewer's Responses to Questions

**Comments to the Author**

1. If the authors have adequately addressed your comments raised in a previous round of review and you feel that this manuscript is now acceptable for publication, you may indicate that here to bypass the “Comments to the Author” section, enter your conflict of interest statement in the “Confidential to Editor” section, and submit your "Accept" recommendation.

Reviewer #1: All comments have been addressed

Reviewer #2: (No Response)

Reviewer #3: (No Response)

2. Is the manuscript technically sound, and do the data support the conclusions?

Reviewer #1: Yes

Reviewer #2: (No Response)

Reviewer #3: Yes

3. Has the statistical analysis been performed appropriately and rigorously? 

Reviewer #1: Yes

Reviewer #2: (No Response)

Reviewer #3: Yes

4. Have the authors made all data underlying the findings in their manuscript fully available?

Reviewer #1: Yes

Reviewer #2: (No Response)

Reviewer #3: Yes

5. Is the manuscript presented in an intelligible fashion and written in standard English?

Reviewer #1: Yes

Reviewer #2: (No Response)

Reviewer #3: Yes

6. Review Comments to the Author

Reviewer #1: (No Response)

Reviewer #2: (No Response)

Reviewer #3: A retrospective observational study was conducted which aimed to develop a predictive model for postpartum endometritis and devise strategies for prevention and control. The significant factors associated with occurrence of endometritis in the postpartum period were: the number of negative finger tests, postpartum hemorrhage, pre-eclampsia, maternity methods, prenatal reproductive tract culture, and uterine exploration. The nomogram model which included these six factors showed promising predictive performance based on the AUCs obtained from both the training and test datasets.

Minor revisions:

1- Express p-values more precisely than “p< 0.05”.

2- Statistical analysis section: Typographical error: Fisher’s exact tests.

3- Model performance evaluation section: Clarify the meaning of “Brier score (0.121 < 0.25) and (0.11 < 0.25)”.

4- Figure 1: Indicate that the data was randomly split by R software.

5- Figure 5: Clearly label the probability scale.

6- Figure 10: Label the x-axis.

7- To assist in the review process, add line numbering to the document.

7. PLOS authors have the option to publish the peer review history of their article (what does this mean?). If published, this will include your full peer review and any attached files.

Reviewer #1: No

Reviewer #2: No

Reviewer #3: No

---

## [Author Response · Author response to Decision Letter 1]

5 Jul 2024

To editors:

1. ACADEMIC EDITOR: 

Please respond to all reviewer comments point by point

clarify the added value of your work over the already known 

Respond to all issues raised by the statistical review.

Response: Thank you for your valuable feedback. Below, we have addressed each of the reviewers' comments individually. Our study focuses on developing and validating a predictive model for maternal postpartum endometritis using three statistical approaches. The risk assessment for maternal postpartum endometritis infection can be easily implemented clinically through a straightforward column chart. Our research represents a significant advancement in endometritis prevention efforts. We have addressed all statistical review concerns comprehensively. Once again, we appreciate your insightful comments.

2. Additional Editor Comments:

Comments from PLOS Editorial Office: We note that one or more reviewers has recommended that you cite specific previously published works in an earlier round of revision. As always, we recommend that you please review and evaluate the requested works to determine whether they are relevant and should be cited. It is not a requirement to cite these works and you may remove them before the manuscript proceeds to publication. We appreciate your attention to this request.

Response: Thank you for your constructive feedback. We have diligently reviewed and assessed the literature referenced in the revised manuscript. These citations have been included to enhance the quality of our work, and we sincerely appreciate your valuable comments once again.

To Reviewers:

Minor revisions:

1- Express p-values more precisely than “p< 0.05”.

Response: Thank you for your constructive feedback. We have enhanced the precision of the P-values presented in Tables 3 and 4, clearly marking them in red for clarity(Revisions to lines 147 and 169). Your comments have been greatly appreciated.

2- Statistical analysis section: Typographical error: Fisher’s exact tests.

Response: Thank you for your constructive feedback. We apologize for any oversight. We have rectified the term "Fisher's exact tests" in the statistical analysis section and highlighted the correction in red(Revisions to line 114). Your comments have been instrumental in improving our work, and we appreciate them once again.

3- Model performance evaluation section: Clarify the meaning of “Brier score (0.121 < 0.25) and (0.11 < 0.25)”.

Response: Thank you for your constructive feedback. We have included an explanation of the Brier score (0.121 < 0.25) and (0.11 < 0.25) in the Model Performance Evaluation section, highlighted in red for clarity(Revisions to lines 195-199). Your comments have been greatly appreciated once again. 

4- Figure 1: Indicate that the data was randomly split by R software.

Response: Thank you for your constructive feedback. We have illustrated in Figure 1 how the data is randomly split using the sample function in R software(Revisions to line 92). Please refer to the newly uploaded Figure 1 for details. Your comments have been sincerely appreciated once again.

5- Figure 5: Clearly label the probability scale.

Response: Thank you for your constructive feedback. We have enhanced Figure 5 by clearly labeling the probability scales(Revisions to line 205). Please refer to the updated Figure 5 for details. Your comments are greatly appreciated once again.

6- Figure 10: Label the x-axis.

Response: Thank you for your constructive feedback. We have now labeled the X-axis in Figure 10(Revisions to lines 248-250). Please refer to the updated Figure 10 for details. Your comments are much appreciated once again.

7- To assist in the review process, add line numbering to the document.

Response: Thank you for your constructive feedback. We have incorporated line numbering into the document. Your comments are sincerely appreciated once again.

---

## [Editor Report · Decision Letter 2]

9 Jul 2024

Development and validation of a predictive model for postpartum endometritis

PONE-D-24-03520R2

Dear Dr. Shao,

We’re pleased to inform you that your manuscript has been judged scientifically suitable for publication and will be formally accepted for publication once it meets all outstanding technical requirements.

Kind regards,

Ahmed Mohamed Maged, MD

Academic Editor

PLOS ONE
---

## [Editor Report · Acceptance letter]

12 Jul 2024

PONE-D-24-03520R2 

PLOS ONE

Dear Dr. Shao, 

I'm pleased to inform you that your manuscript has been deemed suitable for publication in PLOS ONE. Congratulations! Your manuscript is now being handed over to our production team.

Kind regards, 

on behalf of

Professor Ahmed Mohamed Maged 

Academic Editor

PLOS ONE